# Gradual Expansion of a Stent to Prevent Periprocedural Complications after Carotid Artery Stenting for Vulnerable Severe Stenotic Lesions with Intraplaque Hemorrhages: A Retrospective Observational Study

**DOI:** 10.3390/life12010131

**Published:** 2022-01-17

**Authors:** Takahisa Mori, Kazuhiro Yoshioka, Yuhei Tanno, Shigen Kasakura

**Affiliations:** Department of Stroke Treatment, Shonan Kamakura General Hospital, Kamakura City 247-8533, Kanagawa, Japan; y.kazuhiro12@icloud.com (K.Y.); yxip01@icloud.com (Y.T.); kasakura@med.kitasato-u.ac.jp (S.K.)

**Keywords:** carotid artery stenting, intraplaque hemorrhage, periprocedural complications, post-stenting angioplasty, vulnerable lesion

## Abstract

Vulnerable lesions with intraplaque hemorrhages are associated with a high incidence of complications following carotid artery stenting (CAS). CAS for vulnerable lesions has not been established; therefore, we gradually expand stents in such patients. This study aimed to compare the incidences of complications between gradual-expansion CAS for vulnerable lesions and standard CAS for non-vulnerable lesions. For gradual-expansion CAS, we used 3.0 or 4.0 mm balloons for minimal luminal diameters (MLDs) <2.0 or ≥2.0 mm, respectively, for pre-stenting angioplasty (SA) and did not overinflate them. By contrast, for standard CAS, we used a 4.0 mm balloon and overinflated it to 4.23 mm. A closed-cell stent was deployed, and post-SA was not performed in both groups. We evaluated the MLD before and minimal stent diameter (MSD) immediately after CAS, as well as periprocedural complications of combined stroke, death, and myocardial infarction within 30 days after CAS. In the vulnerable and non-vulnerable groups, 30 and 38 patients were analyzed, the MLDs were 0.76 and 0.96 mm before CAS, the MSDs were 2.97 mm and 3.58 mm after CAS, and the numbers of complications were 0 and 1, respectively. Gradual-expansion CAS for vulnerable lesions was as safe as standard CAS for non-vulnerable lesions.

## 1. Introduction

Previous studies have reported that vulnerable carotid lesions with intraplaque hemorrhage (IPH) are associated with ischemic strokes [1], and plaque characteristics are essential in evaluating the risk of ischemic events [2]. Plaque imaging has been developed, and carotid plaque lesions that show hyperintensities on magnetic resonance (MR) images are considered as lesions with IPH [3]. Appropriate surgical treatment is necessary for lesions with IPH [4]. Carotid artery stenting (CAS) is an alternative to carotid endarterectomy (CEA) and is applied in CEA high-risk patients [5]. However, previous studies have reported that lesions with IPH are correlated with higher incidences of a composite outcome of perioperative stroke, death, and myocardial infarction (MI) during or after CAS than those without IPH [6,7]. If carotid lesions are considered vulnerable, CEA high-risk patients should not undergo CAS. To schedule CAS for vulnerable lesions with IPH, it must be established that perioperative complications are minimized and that the safety and feasibility of CAS are similar for vulnerable and non-vulnerable lesions. Only if the CAS risk for vulnerable lesions is the same as that for non-vulnerable lesions can patients with vulnerable lesions be considered to undergo CAS.

Carotid artery plaque imaging with MR imaging (MRI) became available in our institution in 2015, and lesions with IPH were identified; therefore, we implemented gradual-expansion CAS to prevent ruptures due to IPH and reduce periprocedural complications, whereas we continued to perform standard CAS in our institution for lesions without IPH. Gradual-expansion CAS was defined as pre-stenting angioplasty (SA) using a 3.0 mm- or 4.0 mm-diameter balloon catheter without balloon overinflation, deployment of a closed-cell stent without post-SA, and subsequent spontaneous gradual expansion of the implanted stent over a few months. Standard CAS was defined as pre-SA using a 4.0 mm-diameter balloon catheter with balloon overinflation, deployment of a closed-cell stent without post-SA, and subsequent spontaneous expansion of the implanted stent over a few months. Therefore, this retrospective observational study aimed to investigate the safety and short-term clinical and angiographic outcomes of gradual-expansion CAS for severe stenotic vulnerable lesions with IPH compared with our standard CAS for lesions without IPH.

## 2. Materials and Methods

We included patients who underwent (1) elective CAS for lesions with a carotid artery stenosis rate (CASr) ≥ 70% from January 2015 to June 2018 and (2) T1-weighted black blood (T1WBB) MRI for carotid plaque imaging before CAS and diffusion-weighted imaging (DWI) for ischemic lesions before and after CAS. Elective CAS was defined as scheduled CAS in asymptomatic patients or scheduled CAS in patients who experienced the last ischemic attack 30 or more days previously. The CASr was measured according to the North American Symptomatic Carotid Endarterectomy Trial (NASCET) criteria [8]. During the study period, CAS was indicated in symptomatic patients with a CASr ≥ 50% on angiography or asymptomatic patients with a CASr ≥ 60% on angiography according to previous study criteria [9]. However, for analyzing clinical and angiographic outcomes after CAS for severe stenotic lesions with and without IPH in not acute or subacute ischemic stages, we excluded from our analysis patients who (1) underwent CAS within 29 days of the last ischemic attack, (2) had a history of ipsilateral CAS, (3) had a contraindication of MRI, (4) did not undergo CAS due to failure of a stent introduction, (5) underwent CAS for lesions with a CASr < 70%, or (6) underwent CAS with an open-cell stent (Figure 1).

### 2.1. Evaluation

We evaluated patients’ baseline characteristics; plaque relative signal intensity (rSI) on T1WBB MRI; minimal luminal diameter (MLD) before CAS; CASr before and after CAS; lesion laterality; CAS procedural time; middle cerebral artery (MCA) rSI on MR angiography (MRA) before and after CAS; minimal stent diameter (MSD) immediately after and four months after CAS; the presence or absence of new ischemic lesions on DWI (binary analysis) and the absolute number of new ischemic lesions on DWI per patient after CAS (count analysis); the number of hospital days; and any symptomatic stroke, death, or MI within 30 days after CAS. We measured MLD and MSD according to the NASCET criteria [8] for digital subtraction angiography (DSA). Moreover, we assessed the ultrasonographic peak systolic velocity (PSV) before and after CAS. All data were collected prospectively during the study period. We used the radiologists’ reports regarding rSI based on MRI or MRA, ultrasonographers’ reports regarding carotid ultrasonography, and our medical records (YT, KY, SK) regarding MLD, MSD, and CASr based on DSA. MLD, MSD, and CASr were assessed without knowledge of the rSI determined by an independent radiologist.

### 2.2. Carotid Artery Plaque Imaging

We performed T1WBB MRI using a 3.0 T MRI machine (Achieva 3.0T X Quasar, Philips Medical Systems, Tokyo, Japan) equipped with a 16-channel sensitivity-encoding neurovascular coil. The T1WBB MRI protocol was fat-suppressed (FS) three-dimensional (3D) volume isotropic turbo spin-echo acquisition imaging. The parameters for the imaging sequences were as follows: repetition time (TR)/echo time (TE), 500 ms/shortest; flip angle (FA), 90°; matrix, 512 × 304; slice thickness, 1.2 mm; and field of view (FOV), 200 mm. Patients underwent FS 3D T1WBB MRI the day before CAS. 

### 2.3. Definition of Lesions with IPH

The plaque and proximal sternocleidomastoid muscle signal intensities were measured on axial FS 3D T1WBB MRI at a workstation (Zioworkstation2, Ziosoft, Inc., Tokyo, Japan). Lesions with IPH were defined as lesions showing hyperintensity on FS 3D T1WBB MRI, and hyperintensity was defined as an rSI ≥ 1.40 (Figure 2). The plaque rSI was calculated using the following formula: (plaque SI)/(sternocleidomastoid muscle SI) [10]. 

### 2.4. MRA and DWI 

We performed MRA and DWI using a 3.0 T MRI machine, the same as used for FS 3D T1WBB MRI, equipped with an eight-channel sensitivity-encoding head coil. 3D time-of-flight MRA was performed using a 3D fast field-echo sequence. The DWI protocol was echo-planar imaging under the following conditions: TR, 4285 ms; TE, 65.0 ms; slice thickness, 5 mm; slice gap, 1.0 mm; b value, 1000 s/mm^2^; and FOV, 220 mm. Baseline DWI was performed the day before CAS in all patients. A second DWI was performed between 24 and 48 h after CAS, and only new lesions in reference to the baseline DWI were considered new ischemic lesions due to CAS. 

### 2.5. Extremely Severe Carotid Stenosis with Probable High Oxygen Extraction Fraction

A reduced MCA SI on MRA ipsilateral to severe carotid stenosis may be a feasible index to identify patients at high risk of hyperperfusion syndrome after CAS [11,12,13]. We measured the SI on the bilateral MCA M1 segments on MRA and calculated rSI according to a previously reported method [13]. Extremely severe carotid stenosis (ex-CS) with probable high oxygen extraction fraction is defined as severe carotid stenosis of an MLD ≤ 1.06 mm combined with ipsilateral MCA rSI ≤ 0.89 [13], and patients with ex-CS are at high risk of hyperperfusion syndrome following CAS [11,13].

### 2.6. CAS Procedure

We performed CAS through transbrachial catheterization under local anesthesia [14]. A 6-Fr (0.088-inch inner diameter) guiding sheath of 90 cm in length (MSK-guide 7.5 × 90; Medikit, Tokyo, Japan) was positioned in the affected common carotid artery. All patients received 5000 U of heparin intravenously when the 6-Fr guiding sheath was introduced into the brachial artery. We used a filter (Spider FX, Medtronic, Tokyo, Japan) as the embolic distal protection device and administered an additional 5000 U of heparin immediately before the filter device was opened. Pre-SA was performed with the semi-compliant balloon catheter (Shiden, Kaneka Medix, Osaka, Japan), which remained inflated for 30 s unless neurological symptoms occurred. In patients with an ex-CS, the balloon for pre-SA was not overinflated to prevent hyperperfusion syndrome [11]. A stent (Carotid WALLSTENT, Boston Scientific, Tokyo, Japan) was subsequently deployed over the residual stenosis. Post-SA of the residual stent stenosis was not performed. CAS procedural time was defined as the time of staying in the catheterization laboratory. CAS without post-SA was reported to prevent embolic complications [15,16]. Post-SA is regarded as the riskiest part of CAS procedures [17]. Furthermore, long-term spontaneous dilatation of a self-expanding stent after CAS was reported [18]. Therefore, for our standard CAS, a closed-cell stent was used with pre-SA and without post-SA [11]. A closed-cell stent decreases embolic complications by covering a higher percentage of the vascular wall within the stented region than an open-cell stent, thereby avoiding dislodgement of the plaque while the stent expands [19]. Compared with the open-cell stent, the closed-cell stent can slowly self-expand because the radial force is weaker in the closed-cell stent than in the open-cell stent [19,20].

Patients with IPH (IPH group) underwent gradual-expansion CAS. For pre-SA, we used a balloon catheter 3.0 mm in diameter for lesions with MLD < 2.0 and inflated the balloon to 3.0 mm or a semi-compliant balloon catheter 4.0 mm in diameter for lesions with MLD ≥ 2.0 mm and inflated the balloon to 4.0 mm. We did not overinflate them. Subsequently, a closed-cell stent was deployed (Figure 3). 

Patients without IPH (non-IPH group) underwent standard CAS in our institution. A balloon catheter 4.0 mm in diameter for pre-SA was used and overinflated up to 4.23 mm. However, in patients with an ex-CS, the 4.0 mm balloon for pre-SA was not overinflated to prevent hyperperfusion syndrome [11]. Subsequently, a closed-cell stent was deployed (Appendix A). 

### 2.7. DSA Investigation after CAS

As reported previously, we expected the stent to self-expand over a few months. However, some stents fail to expand. Therefore, DSA was performed at four months (ranging from three to six months after CAS) to assess the stent diameter accurately, as reported in previous studies (Figure 3) [21,22]. Residual stenosis was defined as having an MSD < 3.0 mm at four months. The MSD was measured on the lateral view image immediately after and four months after CAS (Figure 4). When patients rejected DSA at four months, carotid ultrasonography was performed to measure the MSD. 

### 2.8. Management before and after CAS 

The patients started to receive dual antiplatelet therapy (DAPT) consisting of clopidogrel (75 mg/day) and cilostazol (100 mg/day) at least seven days before CAS and continued DAPT until 3–6 months of angiographic investigation. Cilostazol was used for DAPT to prevent restenosis [23,24]. The day before CAS, patients began taking the following sedative drugs: 2.5 g of TSUMURA Yokukansan (Kampo, a Japanese herbal medicine) (TJ-54) [25] orally three times a day and 2.0 g of etizolam orally twice a day. Because the sedative effects are expected to decrease oxygen demand and suppress an excessive increase in the cerebral blood flow, TJ-54 was used until seven days after CAS, and etizolam was used until two days after CAS. In addition, antihypertensive drugs were used until seven days after CAS to reduce the systolic blood pressure to <150 mmHg and diastolic blood pressure to <90 mmHg when the blood pressure was elevated after CAS. 

### 2.9. Statistical Analysis 

Non-normally distributed continuous variables are expressed as median and interquartile range. The two groups were compared using the chi-square test or Fisher’s exact test for categorical variables and compared using the Wilcoxon rank-sum test for nonparametric data. The Wilcoxon signed-rank test was used for nonparametric data to compare paired variables. A *p*-value < 0.05 was considered statistically significant. The JMP software (version 16.1; SAS, Cary, NC, USA) was used for the analysis. 

## 3. Results

A total of 167 patients underwent scheduled CAS during the study period. Among them, 68 patients met our inclusion criteria. There were 30 patients in the IPH group and 38 in the non-IPH group (Figure 1). There were no differences in vascular risk factors, the number of ex-CS, or the number of statin users (Table 1). Before CAS, no differences were noted in the MLD, CASr, MCA rSI, PSV, or the number of patients with an ex-CS between the two groups (Table 2). In the IPH group, 20 patients (66.7%) underwent pre-SA with a 3.0 mm balloon catheter, and ten patients underwent pre-SA with a 4.0 mm balloon catheter. Immediately after CAS, the MLD increased, whereas the CASr decreased in overall patients, in the IPH group, and in the non-IPH group (all *p* < 0.001) (Appendix A). The MSD was smaller in the IPH group than in the non-IPH group (*p* = 0.003), and the CASr was larger in the IPH group than in the non-IPH group (*p* = 0.001) (Table 2). The MCA rSI increased (*p* < 0.001), and the PSV decreased (*p* < 0.001) immediately after CAS in overall patients. Likewise, the MCA rSI increased (*p* < 0.001), and the PSV decreased (*p* < 0.001) in both study groups (Appendix A). There was no difference in the MCA rSI before and immediately after CAS between these two groups. By contrast, the PSV immediately after CAS was larger in the IPH group than in the non-IPH group (*p* = 0.02) (Table 2). 

We found no differences in new lesions on DWI (binary or count analysis) after CAS (Table 3). During and after CAS, no transient ischemic attack, cerebral infarction, or intracranial hemorrhage occurred in the two groups. No hyperperfusion syndrome occurred in overall patients. No stroke, death, or MI occurred within 30 days after CAS in the IPH group, whereas one patient died due to heart failure 26 days after CAS in the non-IPH group (Table 3, Figure 1).

At four months after CAS, 66 of 68 patients were investigated, 59 patients underwent DSA, and 7 patients underwent carotid ultrasonography only (Figure 1). DSA demonstrated an increase in MSD at four months and a decrease in CASr compared with those immediately after CAS in overall patients, in the IPH group, and in the non-IPH group (all *p* < 0.001) (Appendix A). There were no differences in the MSD and CASr observed at four months between the two groups (Table 2). However, the MSD change in the IPH group was significantly higher than that in the non-IPH group (*p* = 0.03) (Table 2, Figure 5). In the IPH group (*n* = 29) and non-IPH group (*n* = 33), stents remained stenotic in four and one patients (*p* = 0.160), respectively.

The 3.0 mm balloon had been used for pre-SA in three of the four patients with residual stent stenosis in the IPH group. The five patients with residual stent stenosis underwent balloon angioplasty at four months, and their lesions were sufficiently dilated without any complications.

## 4. Discussion

Our results suggest that gradual-expansion CAS for lesions with IPH was as safe as standard CAS for lesions without IPH, and stents spontaneously self-expanded in the two groups. In addition, no combined complications occurred within 30 days following CAS in the IPH group. Our results also suggest that pre-CAS T1WBB MRI and MRA were essential to prevent periprocedural complications following CAS.

We performed transbrachial CAS during the study period; however, transradial CAS has also been carried out in other studies [26]. We attempted gradual-expansion CAS for vulnerable lesions with IPH. However, gradual-expansion procedures are almost identical to standard procedures except for pre-SA 3.0 mm balloon use in vulnerable lesions and overinflation to 4.23 mm of pre-SA in lesions without IPH. Therefore, standard CAS for vulnerable lesions may be as safe as gradual-expansion CAS, although we have not evaluated the safety of standard CAS for vulnerable lesions. Previous studies have reported a higher rate of combined stroke, death, and MI within 30 days following CAS for lesions with IPH (8.1%) than following CAS for lesions without IPH (2.1%) [7]. In our study, no case of combined stroke, death, and MI within 30 days occurred following gradual-expansion CAS in the IPH group, and one case occurred following standard CAS in the non-IPH group. In our study, the binary analysis showed that new DWI lesions were found in 60.0% and 68.4% of the patients in the IPH and non-IPH groups, respectively. Those incidence rates may be high compared with 50.0% in a previous report [27], whereas the median count of new DWI lesions was one in our study. Symptomatic ischemic or hemorrhagic complications did not occur in the two groups. A low number of new lesions may be associated with asymptomatic ischemic complications, as the number of new DWI lesions is associated with stroke after CAS [27]. An incidence rate of 6.4% of hemorrhagic transformation was reported, and the perioperative stroke or mortality rate was high in patients with multiple comorbidities in whom CEA had been performed within 20 days [28]. In both our study groups, elective CAS was performed 30 days or more after the last ischemic symptom to prevent the hemorrhagic transformation of ischemic infarction. Between immediately after and four months after CAS, the MSD in the IPH group increased more than that in the non-IPH group, indicating the gradual expansion of stents. 

The median age of the study population was 77 years, which was older than the mean age of 72.5 years in the CAS group in a previous study [5]. Nevertheless, no combined complications occurred in the IPH group.

Previous studies have reported not only CAS without post-SA [15] but also CAS without any balloon angioplasty [18,29]. Primary CAS without any angioplasty is simple, and implanted stents continue to expand after deployment. Therefore, primary CAS is a possible treatment option for vulnerable lesions despite the high restenosis rate (12.8%) and high repeat angioplasty rate (14.8%) [18,19]. In our study, 4 of 26 patients (15.4%) in the IPH group underwent repeat balloon angioplasty, and the problem of long-term residual stent-stenosis should be resolved. 

Previous studies have reported the double-balloon protection technique for vulnerable lesions [4,30]. This technique is rational but complicated as a routine procedure. However, gradual-expansion CAS is more feasible and straightforward compared with the double-balloon protection technique. 

Our study used a 3.0 T MRI machine with a neurovascular coil and defined hyperintensity as an rSI ≥ 1.4 in FS 3D T1WBB MRI to identify lesions with IPH. However, previous studies of lesions with IPH used different magnetic field strengths and imaging protocol and parameters for MRI [3,31,32]. A previous study used an rSI ≥ 1.25 to define vulnerable lesions [4]. If MRI is used to identify IPH in CAS or CEA, the standardization of the magnetic field strength, imaging protocol and parameters, and definition of the rSI may be required. 

Gradual-expansion CAS for vulnerable lesions is feasible, and its procedural risk is almost the same as that of CAS for non-vulnerable lesions. Therefore, gradual-expansion CAS may be the first-line CAS option in patients at high risk of CEA.

### Limitations

Our study has several limitations. First, a small number of patients were included, and the study design was a retrospective cohort study, although data were collected prospectively. Second, the appropriate MLD to determine the pre-SA balloon must be determined, although we selected an MLD of 2.0 mm as the index to distinguish the use of a 3.0 mm balloon from that of a 4.0 mm balloon. Third, we used the same balloon catheters, filter devices, and closed-cell stents in the IPH group. Our results cannot be generalized for other types of balloon catheters, filter devices, or closed-cell stents. Therefore, the appropriate balloon catheter and balloon size for pre-SA and appropriate stent must be determined. Fourth, the follow-up time was short. One- or two-year follow-up is necessary to investigate the long-term effects of spontaneous dilatation in the gradual-expansion CAS group. Finally, a prospective study with a large number of patients is required to evaluate CAS for lesions with IPH to confirm our results. 

## 5. Conclusions

Gradual-expansion CAS for vulnerable lesions was feasible and as safe as standard CAS for lesions without IPH. Symptomatic complications after gradual-expansion CAS did not occur. Therefore, gradual-expansion CAS for vulnerable carotid lesions may be the first-line CAS strategy in CEA high-risk patients.

## Figures and Tables

**Figure 1 life-12-00131-f001:**
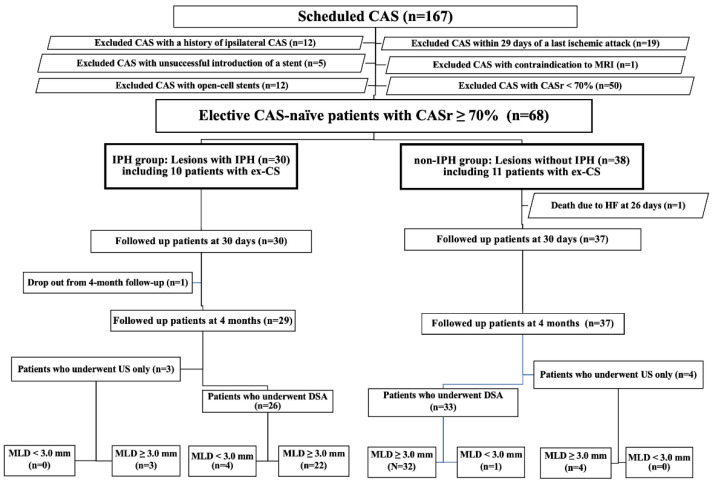
Flowchart of the patient selection process for the analysis. CAS, carotid artery stenting; CASr, carotid artery stenosis rate; DSA, digital subtraction angiography; ex-CS, extremely high-grade carotid stenosis; HF, heart failure; IPH, intraplaque hemorrhage; MLD, minimal luminal diameter; MRI, magnetic resonance imaging; US, ultrasonography.

**Figure 2 life-12-00131-f002:**
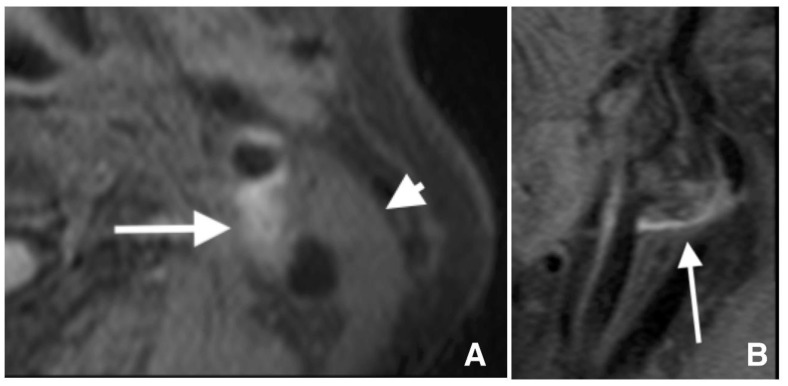
T1-weighted black blood magnetic resonance imaging reveals a high signal intensity of 1533 in carotid lesions (arrows) and a signal intensity of 636 in the sternocleidomastoid muscle (arrowhead) in axial (**A**) and sagittal (**B**) images, indicating a relative signal intensity of 2.44.

**Figure 3 life-12-00131-f003:**
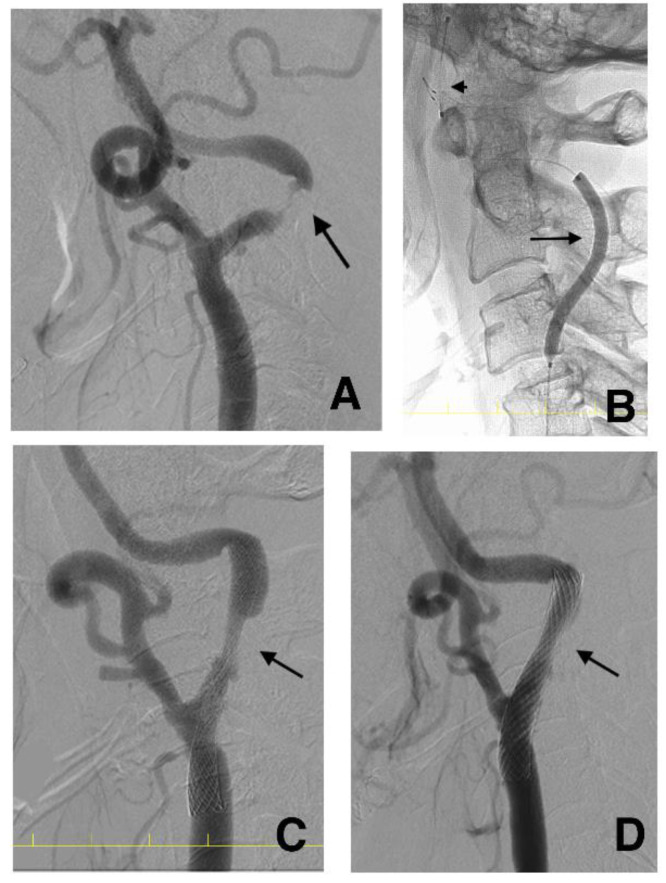
Serial angiograms before (**A**), immediately after (**C**), and at four months after (**D**) carotid artery stenting (CAS) and the balloon (arrow) and filter (arrowhead) during inflation (**B**) in the same case as in Figure 2. The minimal luminal diameter is 0.26 mm (**A**: arrow). The semi-compliant balloon flexibly dilates the lesion along the artery (**B**: arrow). The minimal stent diameter is 2.69 mm after CAS (**C**: arrow), and the minimal stent diameter at four months is 3.97 mm (**D**: arrow). CAS, carotid artery stenting.

**Figure 4 life-12-00131-f004:**
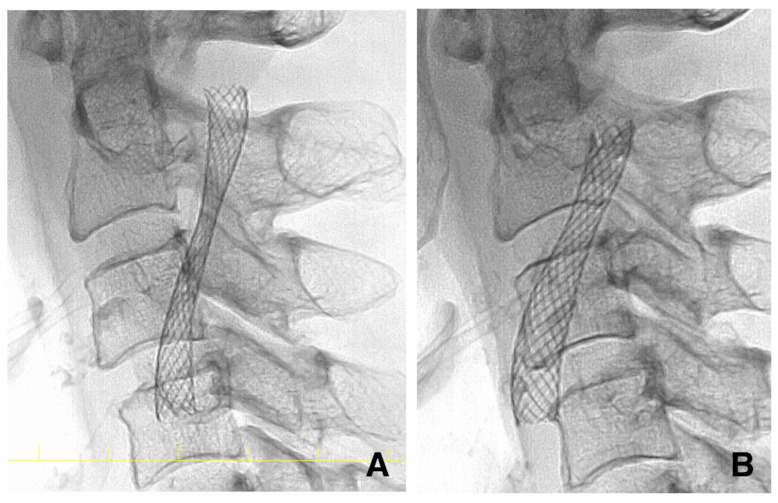
Pictures of the stent in the same case as in Figure 2. (**A**) The stent is stenotic immediately after carotid artery stenting. (**B**) The stent has expanded spontaneously at four months.

**Figure 5 life-12-00131-f005:**
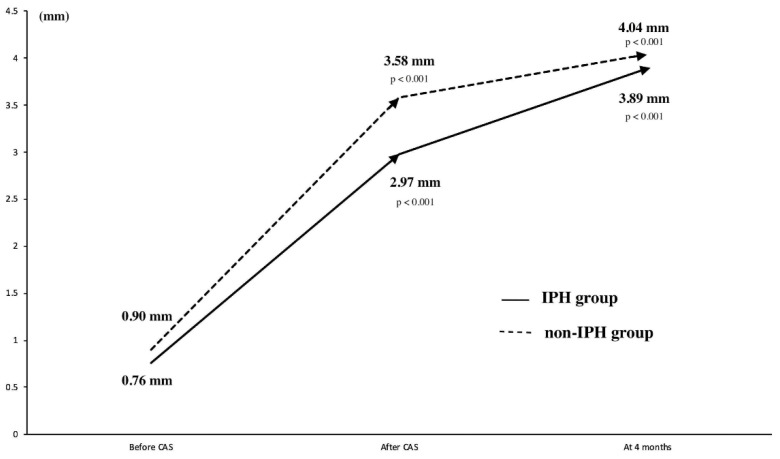
Serial changes in luminal diameters before, immediately after, and four months after CAS. The solid line shows changes in the diameters in the IPH group. The dotted line shows changes in the diameters in the non-IPH group. CAS, carotid artery stenting; IPH, intraplaque hemorrhage.

**Table 1 life-12-00131-t001:** Patient characteristics.

Groups	Overall	IPH Group	Non-IPH Group	
	*n* = 68	*n* = 30	*n* = 38	
CAS Procedure		GE	Standard	*p*-Value
Plaque rSI on T1WBB MRI,	1.32 (1.11, 1.74)	1.84 (1.6, 2.0)	1.1 (1.0, 1.2)	<0.001
Age, years	77 (72, 81)	78 (71, 81)	77 (73, 80.5)	0.92
Male (sex), *n* (%)	59 (86.8%)	28 (90.3%)	38 (77.6%)	0.13
BMI on admission, kg/m^2^	23.2 (21.0, 24.6)	23.4 (20.7, 24.4)	23.1 (21.0, 25.1)	0.58
TCHO on admission, mmol/L	4.3 (3.7, 4.7)	4.2 (3.7, 4.6)	4.3 (3.7, 5.0)	0.99
LDL on admission, mmol/L	2.2 (1.8, 2.5)	2.1 (1.6, 2.6)	2.2 (2.0, 2.4)	0.44
HDL on admission, mmol/L	1.3 (1.1, 1.7)	1.5 (1.2, 1.8)	1.2 (1.1, 1.6)	0.06
TG on admission, mmol/L	1.3 (0.9, 1.9)	1.3 (0.8, 1.6)	1.3 (0.9, 1.9)	0.45
Glucose on admission, mmol/L	6.2 (5.6, 7.4)	6.7 (5.7, 8.1)	6.0 (5.5, 7.2)	0.20
HbA1c on admission, %	5.9 (5.7, 6.3)	5.8 (5.7, 6.3)	5.9 (5.7, 6.3)	0.52
Symptomatic, *n* (%)	35 (51.5%)	19 (63.3%)	16 (42.1%)	0.08
Statin users, *n* (%)	36 (52.9%)	17 (56.7%)	19 (50.0%)	0.58
History of hypertension, *n* (%)	62 (91%)	28 (93.3%)	35 (92.1%)	0.84
History of diabetes mellitus, *n* (%)	33 (49%)	16 (53.3%)	17 (44.7%)	0.48
Number of hospital days, days	4 (4, 5)	5 (4, 5)	4 (4, 5)	0.23

All values except for categorical data are represented as median (interquartile range). BMI, body mass index; CAS, carotid artery stenting; GE, gradual expansion; HDL, high-density lipoprotein cholesterol; IPH, intraplaque hemorrhage; LDL, low-density lipoprotein cholesterol; *n*, number; *p*, probability; rSI, relative signal intensity; T1WBB MRI, T1-weighted black blood magnetic resonance imaging; TCHO, total cholesterol; TG, triglyceride.

**Table 2 life-12-00131-t002:** Digital subtraction and magnetic resonance angiographic information before and after carotid artery stenting.

Groups	Overall	IPH Group	Non-IPH Group	
	*n* = 68	*n* = 30	*n* = 38	
CAS Procedure		GE	Standard	*p*-Value
MCA slow flow before CAS, *n* (%)	35 (51.5%)	15 (50.0%)	20 (52.6%)	0.83
MCA rSI < 0.9	39 (56.3%)	16 (53.3%)	23 (60.5%)	0.55
MLD < 1.0 mm	41 (60.3%)	19 (63.3%)	22 (57.9%)	0.65
CASr ≥ 80%	39 (57.3%)	17 (56.7%)	22 (57.9%)	0.92
ex-CS	21 (30.8%)	10 (33.3%)	11 (28.9%)	0.70
MCA rSI before CAS	0.87 (0.74, 0.98)	0.88 (0.74, 0.98)	0.87 (0.74, 0.96)	0.54
MCA rSI after CAS	1.01 (0.92, 1.06)	1.02 (0.97, 1.06)	0.98 (0.89, 1.07)	0.33
PSV before CAS, cm/s	281 (210, 348)	271 (215, 337)	293 (178, 361)	0.69
PSV after CAS, cm/s	83 (67.2, 113.8)	97.6 (70.0, 147)	80.1 (64.9, 91.2)	0.02
CASr before CAS, %	80.9 (76.9, 86.7)	81.2 (77.4, 88.1)	80.7 (76.5, 86.0)	0.68
CASr after CAS, %	29.2 (18.6, 38.8)	37.5 (26.6, 44.6)	26.4 (15.9, 34.2)	0.001
CASr at four months, %	26.0 (16.5, 33.5)	27.7 (18.5, 40.7)	24.3 (14.2, 32.2)	0.28
Change of CASr over four months, %	−11.6 (−18.0, −8.46)	−12.6 (−20.2, −9.0)	−10.9 (−17.1, −6.3)	0.14
MLD before CAS, mm	0.85 (0.54, 1.09)	0.76 (0.54, 1.05)	0.90 (0.53, 1.14)	0.74
MSD after CAS, mm	3.42 (2.82, 3.73)	2.97 (2.62, 3.54)	3.58 (3.13, 3.86)	0.003
MSD at four months, mm	3.97 (3.46, 4.47)	3.89 (3.22, 4.75)	4.04 (3.57, 4.33)	0.61
Change in MSD over four months, mm	0.76 (0.46, 1.02)	0.90 (0.62, 1.08)	0.58 (0.31, 0.94)	0.03

All values except for categorical data are represented as median (interquartile range). CAS, carotid artery stenting; CASr, carotid artery stenosis rate; ex-CS, extremely high-grade carotid stenosis; GE, gradual expansion; IPH, intraplaque hemorrhage; MCA, middle cerebral artery; MLD, minimal luminal diameter; MSD, minimal stent diameter; *n*, number; *p*, probability; PSV, peak systolic velocity; rSI, relative signal intensity.

**Table 3 life-12-00131-t003:** Complications after carotid artery stenting.

Groups	IPH Group	Non-IPH	
	*n* = 30	*n* = 38	
CAS Procedure	GE	Standard	*p*-Value
New ischemic lesions on DWI (binary analysis), *n* (%)	18 (60.0%)	26 (68.4%)	0.47
Number of new lesions on DWI per patient (count analysis), median (IQR)	1 (0, 3)	1 (0, 3)	0.40
Transient ischemic attack, *n*	0	0	1
Symptomatic cerebral infarction after CAS, *n*	0	0	1
Triad of CHS after CAS, *n*	0	0	1
Intracranial hemorrhage after CAS, *n*	0	0	1
Transient delirium, *n* (%)	1 (3.3%)	0	0.44
Hypotension after CAS, *n* (%)	0	2 (5.3%)	0.50
Pseudoaneurysm in the brachial artery, *n* (%)	0	2 (5.3%)	0.50
Stroke, MI, or death within 30 days after CAS, *n* (%)	0	1 (2.6%)	1

CAS, carotid artery stenting; CHS, cerebral hyperperfusion syndrome; DWI, diffusion-weighted imaging; GE, gradual expansion; IPH, intraplaque hemorrhage; IQR, interquartile range; MI, myocardial infarction; *n*, number; *p*, probability.

## Data Availability

The datasets generated and/or analyzed during the current study are available from the corresponding author upon reasonable request.

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
