# Peer review of "Gradual Expansion of a Stent to Prevent Periprocedural Complications after Carotid Artery Stenting for Vulnerable Severe Stenotic Lesions with Intraplaque Hemorrhages: A Retrospective Observational Study"

_life, 2022, doi:10.3390/life12010131_

Round 1

Reviewer 1 Report

This study aimed to compare the incidence of complications between gradual-expansion CAS for vulnerable lesions and standard CAS for non-vulnerable lesions. The study findings showed that gradual-expansion CAS for vulnerable lesions was as safe as standard CAS for non-vulnerable lesions. Overall, this study used appropriate statistical methods to draw study conclusions. Here are my suggestions:

  1. In Methods, please describe the measurement methods for MLD and MSD in detail. In addition, it should be added which authors measured MLD, MSD, and rSI and whether they were blinded.
  2. Luminal diameter seems to be one of the important outcomes of this study. Therefore, inter- and intra-rater reliability for MLD/MSD is required.
  3. The flowchart, which is Figure 1, is partially cropped.
  4. In the introduction, distinguish between the two by clarifying the definitions of gradual expansion CAS and standard CAS. In addition, it should be specifically described why it is important to prove the non-inferiority of gradual expansion CAS. Moreover, please explain why gradual expansion CAS is more desirable in IPH patients.
  5. In the discussion, the rationale and reason why gradual expansion CAS is suggested as a first-line treatment option rather than standard CAS in patients with IPH are very insufficient. These should be supplemented.

Author Response

This study aimed to compare the incidence of complications between gradual-expansion CAS for vulnerable lesions and standard CAS for non-vulnerable lesions. The study findings showed that gradual-expansion CAS for vulnerable lesions was as safe as standard CAS for non-vulnerable lesions. Overall, this study used appropriate statistical methods to draw study conclusions. Here are my suggestions:

Q1-1)      In Methods, please describe the measurement methods for MLD and MSD in detail.

Response 1-1)       Thank you for your valuable comments.

To address this point, we have added a sentence to subsection 2.1. Evaluation of the Methods section as follows:

“We measured MLD and MSD according to the NASCET criteria [8] for digital subtraction angiography (DSA).”

Q1-2)      In addition, it should be added which authors measured MLD, MSD, and rSI and whether they were blinded.

Response 1-2)        We describe now who measured MLD, MSD, and rSI in subsection 2.1. Evaluation of the Methods section as follows:

“We used the radiologists’ reports regarding rSI based on MRI or MRA, ultrasonographers’ reports regarding carotid ultrasonography, and our medical records (YT, KY, SK) regarding MLD, MSD, and CASr based on DSA.”

We have revised the sentence parts in red. MLD, MSD, and CASr were assessed without knowledge of the rSI determined by an independent radiologist.

Q2)         Luminal diameter seems to be one of the important outcomes of this study. Therefore, inter-and intra-rater reliability for MLD/MSD is required.

Response 2) Thank you for your insightful comments and pertinent suggestions.

We agree that inter-and intra-rater reliability for MLD and MSD are important parameters; however, the NASCET criteria [8] for their measurement are commonly used in the field of CEA and CAS research. In addition, we used the MLD and MSD described in our medical records. Therefore, we have not additionally determined the inter-and intra-rater reliability using Cohen's Kappa or intraclass correlation coefficients.

Q3)         The flowchart, which is Figure 1, is partially cropped.

Response 3) Thank you for this important comment. We have revised the layout of Figure 1 in the manuscript to prevent automatic cropping.

 Q4-1)      In the introduction, distinguish between the two by clarifying the definitions of gradual expansion CAS and standard CAS.

Response 4-1) Thank you for this thoughtful suggestion. Accordingly, we have added the following sentences to the Introduction on page 2:

“Gradual-expansion CAS was defined as pre-stenting angioplasty (SA) using a 3.0- or 4.0 mm-diameter balloon catheter without balloon overinflation, deployment of a closed-cell stent without post-SA, and subsequent spontaneous gradual expansion of the implanted stent over a few months. Standard CAS was defined as pre-SA using a 4.0 mm-diameter balloon catheter with balloon overinflation, deployment of a closed-cell stent without post-SA, and subsequent spontaneous expansion of the implanted stent over a few months.”

Q4-2)      In addition, it should be specifically described why it is important to prove the non-inferiority of gradual expansion CAS.

Response 4-2)       We have added and revised the sentences in the Introduction section on page 1 as follows:

“If carotid lesions are considered vulnerable, CEA high-risk patients should not undergo CAS. To schedule CAS for vulnerable lesions with IPH, it must be established that perioperative complications are minimized and that the safety and feasibility of CAS are similar for vulnerable and non-vulnerable lesions. Only if the CAS risk for vulnerable lesions is the same as that for non-vulnerable lesions, patients with vulnerable lesions can be considered to undergo CAS.”

Q4-3)      Moreover, please explain why gradual expansion CAS is more desirable in IPH patients.

Response 4-3)       We have revised the sentence in the Introduction section on page 2 as follows:

“…, we implemented gradual-expansion CAS to prevent ruptures due to IPH and reduce periprocedural complications,…”

Q5)         In the discussion, the rationale and reason why gradual expansion CAS is suggested as a first-line treatment option rather than standard CAS in patients with IPH are very insufficient. These should be supplemented.

Response 5) Thank you very much for your comments and constructive suggestions. We have revised the final sentences in the Discussion section on page 10 as follows:

“Gradual-expansion CAS for vulnerable lesions is feasible, and its procedural risk is almost the same as that of CAS for non-vulnerable lesions. Therefore, gradual-expansion CAS may be the first-line CAS option in patients at high risk of CEA.”

Reviewer 2 Report

In this paper the authors present a study to determine the feasibility of gradual stent expansion for CAS procedures as alternatives to standard therapy.  The study includes two aspects, those related to the clinic (therapeutic procedure, treatment complications) and those related to imaging exploration.

Therefore, more conclusions can be added regarding:

- procedures performed on the types of vascular lesions,

- the evolution pre- and poststenosis imaging aspects,

- the types of complications after stenting of the carotid artery  visualized only imagistically or with few clinical consequences.

Their conclusion was that gradual - expansion CAS for vulnerable lesions was as safe as standard CAS for non-vulnerable lesions. 

The article is well written and documented. I recommend it for publication with minor touch ups. 

Author Response

In this paper the authors present a study to determine the feasibility of gradual stent expansion for CAS procedures as alternatives to standard therapy. The study includes two aspects, those related to the clinic (therapeutic procedure, treatment complications) and those related to imaging exploration.

The article is well written and documented. I recommend it for publication with minor touch ups.

Therefore, more conclusions can be added regarding:

- procedures performed on the types of vascular lesions, - the evolution pre- and poststenosis imaging aspects,

- the types of complications after stenting of the carotid artery visualized only imagistically or with few clinical consequences.

Their conclusion was that gradual - expansion CAS for vulnerable lesions was as safe as standard CAS for non-vulnerable lesions.

Response ) We appreciate your valuable suggestions. Based on your comments, we have added the following sentences to the Conclusions section:

“Gradual-expansion CAS for vulnerable lesions was feasible and as safe as standard CAS for lesions without IPH. Symptomatic complications after gradual-expansion CAS did not occur. Therefore, gradual-expansion CAS for vulnerable carotid lesions may be the first-line CAS strategy in CEA high-risk patients.”
